# Learning Neural Random Fields with Inclusive Auxiliary Generators

## Abstract

Neural random fields (NRFs), which are defined by using neural networks to implement potential functions in undirected models, provide an interesting family of model spaces for machine learning. In this paper we develop a new approach to learning NRFs with inclusive-divergence minimized auxiliary generator - the inclusive-NRF approach, for continuous data (e.g. images), with solid theoretical examination on exploiting gradient information in model sampling. We show that inclusive-NRFs can be flexibly used in unsupervised/supervised image generation and semi-supervised classification, and empirically to the best of our knowledge, represent the best-performed random fields in these tasks. Particularly, inclusive-NRFs achieve state-of-the-art sample generation quality on CIFAR-10 in both unsupervised and supervised settings. Semi-supervised inclusive-NRFs show strong classification results on par with state-of-the-art generative model based semi-supervised learning methods, and simultaneously achieve superior generation, on the widely benchmarked datasets - MNIST, SVHN and CIFAR-10.

## 1 Introduction

One of the core research problems in machine learning is learning with probabilistic models, which can be broadly classified into two classes - directed and undirected[1] (Koller & Friedman, 2009). Significant progress has been made recently on learning with deep generative models (DGMs), which generally refer to models with multiple layers of stochastic or deterministic variables. There have emerged a bundle of deep directed generative models, such as variational AutoEncoders (VAEs) (Kingma & Welling, 2014), generative adversarial networks (GANs) (Goodfellow et al., 2014) and so on. In contrast, undirected generative models (also known as random fields (Koller & Friedman, 2009), energy-based models (LeCun et al., 2006)) received less attention with slow progress. This is presumably because fitting undirected models is more challenging than fitting directed models. In general, calculating the log-likelihood and its gradient is analytically intractable, because it involves the expectation with respect to (w.r.t.) the model distribution.

In this paper, we aims to advance the learning of neural random fields (RFs) which use neural networks with multiple (deterministic) layers to define the potential function[2] $u_\theta(x)$ over observation $x$ with parameter $\theta$. The probability distribution $p_\theta(x) \propto \exp(u_\theta(x))$ is then defined by normalizing the exponentiated potential function. This type of RFs has been studied several times in different contexts, once called deep energy models (DEMs) (Ngiam et al., 2012; Kim & Bengio, 2016), descriptive models (Xie et al., 2016), generative ConvNet (Dai et al., 2014), neural random field language models (Wang & Ou, 2017). For convenience, we refer to such models as **neural random fields (NRFs)** in general.

An important method of maximum likelihood (ML) learning of random fields is called stochastic maximum likelihood (SML) (Younes, 1989), which approximates the model expectations by Monte Carlo sampling for calculating the gradient. A recent progress in learning NRFs as studied in Kim & Bengio (2016); Xie et al. (2016); Wang & Ou (2017); Kuleshov & Ermon (2017) is to

---

[1]An easy way to tell an undirected model from a directed model is that an undirected model involves the normalizing constant (also called the partition function in physics), while the directed model is self-normalized.

[2]Note that compared to modeling with multiple deterministic layers, modeling with multiple stochastic layers, e.g. deep Boltzmann machines (DBMs) (Salakhutdinov & Hinton, 2009), presents much greater challenge for model learning and thus yields inferior performance. This is observed in both directed and undirected models.

pair the target random field $p_\theta$ with an auxiliary directed generative model (often called generator) $q_\phi(x)$ parameterized by $\phi$, which approximates sampling from the target random field. Learning is performed by maximizing the log-likelihood of training data under $p_\theta$ or some bound of the log-likelihood, and simultaneously minimizing some divergence between the target random field $p_\theta$ and the auxiliary generator $q_\phi$. Different learning algorithms differ in the objective functions used in the joint training of $p_\theta$ and $q_\phi$, and thus have different computational and statistical properties (partly illustrated in Figure 1). For example, minimizing the **exclusive-divergence** $KL[q_\phi||p_\theta] \triangleq \int q_\phi \log(q_\phi/p_\theta)$ w.r.t. $\phi$, as employed in Kim & Bengio (2016), involves the intractable entropy term and tends to enforce the generator to seek modes, yielding missing modes. There are also other factors, e.g. modeling discrete or continuous data, different choices of the target RF and the generator, which lead to different algorithms. We leave detailed comparison and connection of our approach with existing studies to section 3 (related work).

In this paper, we propose to use inclusive-divergence minimized auxiliary generators (section 2.1). And particularly for continuous data (e.g. images), we propose to use SGLD (stochastic gradient Langevin dynamics (Welling & Teh, 2011)) and SGHMC (stochastic gradient Hamiltonian Monte Carlo (Chen et al., 2014)) to exploit gradient information in model sampling with solid theoretical examination (section 2.2). The new approach, abbreviated as the **inclusive-NRF** approach, offers some advantages over previous methods. First, minimizing the **inclusive-divergence** $KL[p_\theta||q_\phi] \triangleq \int p_\theta \log(p_\theta/q_\phi)$ w.r.t. $\phi$ avoids the annoying entropy term and tends to drive the generator to cover modes of the target density $p_\theta$. The SGLD/SGHMC sampling further pushes the samples towards the modes of $p_\theta$. Presumably, this helps to produce Markov chains that mix fast between modes and facilitate model learning. Second, the new approach enables us to flexibly use NRFs in unsupervised/supervised image generation and semi-supervised classification (section 2.3), and empirically to the best of our knowledge, represents the best-performed random fields in these tasks.

The main contributions of this paper can be summarized as follows:

- We develop the inclusive-NRF approach, which learns NRFs with inclusive auxiliary generators and particularly for continuous data, exploits gradient information in model sampling with solid theoretical examination.
- Inclusive-NRFs achieve state-of-the-art sample generation quality, measured by both Inception Score (IS) and Frechet Inception Distance (FID). On CIFAR-10, we obtain unsupervised IS 8.28 (FID 20.9) and supervised IS 9.06 (FID 18.1), both using unconditional generation.
- Semi-supervised inclusive-NRFs show strong classification results on par with state-of-the-art DGM-based semi-supervised learning (SSL) methods, and simultaneously achieve superior generation, on the widely benchmarked datasets - MNIST, SVHN and CIFAR-10.

## 2 THE INCLUSIVE-NRF APPROACH

Consider a random field for modeling observation $x$ with parameter $\theta$:

$$p_\theta(x) = \frac{1}{Z(\theta)} \exp[u_\theta(x)] \qquad (1)$$

where $Z(\theta) = \int \exp(u_\theta(x))dx$ is the normalizing constant, $u_\theta(x)$ is the potential function[3] which assigns a scalar value to each configuration of random variable $x$. The general idea of neural random fields (NRFs) is to implement $u_\theta(x) : \mathbb{R}^{d_x} \to \mathbb{R}$, by a neural network, taking the multi-dimensional $x \in \mathbb{R}^{d_x}$ as input and outputting the scalar $u_\theta(x) \in \mathbb{R}$. In this manner, we can take advantage of the representation power of neural networks for RF modeling. It is usually intractable to maximize the data log-likelihood $log p_\theta(\tilde{x})$ for observed $\tilde{x}$, since the gradient involves expectation w.r.t. the model distribution, as shown below:

$$\nabla_\theta \log p_\theta(\tilde{x}) = \nabla_\theta u_\theta(\tilde{x}) - E_{p_\theta(x)}[\nabla_\theta u_\theta(x)] \qquad (2)$$

### 2.1 INTRODUCING INCLUSIVE-DIVERGENCE MINIMIZED AUXILIARY GENERATORS

In this paper, we further develop NRF learning with auxiliary generators. We are mainly concerned with modeling fixed-dimensional continuous observations $x \in \mathbb{R}^{d_x}$ (e.g. images), and choose a

---

[3] Negating the potential function defines the energy function.

---

**Algorithm 1** Learning NRFs with inclusive auxiliary generators

---
**repeat**
    Sampling: Draw a minibatch $\mathcal{M} = \left\{ (\tilde{x}^i, x^i, h^i), i = 1, \cdots |\mathcal{M}| \right\}$ from $\tilde{p}(\tilde{x}) p_\theta(x) q_\phi(h|x)$ (see Algorithm 2);
    Updating:
    Update $\theta$ by ascending: $\frac{1}{|\mathcal{M}|} \sum_{(\tilde{x}, x, h) \sim \mathcal{M}} \left[ \nabla_\theta u_\theta(\tilde{x}) - \nabla_\theta u_\theta(x) \right]$;
    Update $\phi$ by ascending: $\frac{1}{|\mathcal{M}|} \sum_{(\tilde{x}, x, h) \sim \mathcal{M}} \nabla_\phi \log q_\phi(x, h)$;
**until** convergence

---

directed generative model, $q_\phi(x, h) \triangleq q(h) q_\phi(x|h)$, for the auxiliary generator, which specifically is defined as follows[4]:

$$h \sim \mathcal{N}(0, I_h),$$
$$x = g_\phi(h) + \epsilon, \epsilon \sim \mathcal{N}(0, \sigma^2 I_\epsilon), \tag{3}$$

where $g_\phi(h) : \mathbb{R}^{d_h} \to \mathbb{R}^{d_x}$ is implemented as a neural network with parameter $\phi$, which maps the latent code $h$ to the observation space. $I_h$ and $I_\epsilon$ denote the identity matrices, with dimensionality implied by $h$ and $\epsilon$ respectively. Drawing samples from the generator $q_\phi(x, h)$ is simple as it is just ancestral sampling from a 2-variable directed graphical model.

Suppose that data $\mathcal{D} = \{\tilde{x}_1, \cdots, \tilde{x}_n\}$, consisting of $n$ observations, are drawn from the true but unknown data distribution $p_0(\cdot)$. $\tilde{p}(\tilde{x}) \triangleq \frac{1}{n} \sum_{k=1}^{n} \delta(\tilde{x} - \tilde{x}_k)$ denotes the empirical data distribution. Then we formulate the maximum likelihood learning of $p_\theta(x)$ with the inclusive-divergence minimized generator $q_\phi(x)$ as optimizing[5]

$$\begin{cases} \min_\theta KL\left[ \tilde{p}(\tilde{x}) || p_\theta(\tilde{x}) \right] \\ \min_\phi KL\left[ p_\theta(x) || q_\phi(x) \right] \end{cases} \tag{4}$$

The first line of Eq. (4) is equivalent to maximum likelihood training of the target RF $p_\theta$ under the empirical data $\tilde{p}$, which requires sampling from $p_\theta$. Simultaneously, the second line optimizes the generator $q_\phi$ to be close to $p_\theta$ so that $q_\phi$ becomes a good proposal for sampling from $p_\theta$. It can be easily seen that the gradients w.r.t. $\theta$ and $\phi$ (to be ascended) are defined as follows:

$$\begin{cases} \nabla_\theta = E_{\tilde{p}(\tilde{x})}\left[ \nabla_\theta \log p_\theta(\tilde{x}) \right] = E_{\tilde{p}(\tilde{x})}\left[ \nabla_\theta u_\theta(\tilde{x}) \right] - E_{p_\theta(x)}\left[ \nabla_\theta u_\theta(x) \right], \\ \nabla_\phi = E_{p_\theta(x)}\left[ \nabla_\phi \log q_\phi(x) \right] = E_{p_\theta(x) q_\phi(h|x)}\left[ \nabla_\phi \log q_\phi(x, h) \right]. \end{cases} \tag{5}$$

Both lines of Eq. (5) hold, as proved in Proposition 1 in the Supplement. In practice, we calculate noisy gradient estimators, and apply minibatch based stochastic gradient descent (SGD) to solve the optimization problem Eq. (4), as shown in Algorithm 1.

## 2.2 APPLYING SGLD/SGHMC FOR MODEL SAMPLING

In Algorithm 1, we need to draw samples from $p_\theta(x) q_\phi(h|x)$ given current $\theta$ and $\phi$. For continuous observations, SGLD (stochastic gradient Langevin dynamics) (Welling & Teh, 2011) and SGHMC (Stochastic Gradient Hamiltonian Monte Carlo) (Chen et al., 2014) sampling provide mechanisms for exploiting (stochastic) gradients of the target density $p_\theta(x) q_\phi(h|x)$, enabling efficient exploration of the state space. We take the theoretical results about SGLD from Teh et al. (2016) and SGHMC from Chen et al. (2014), which are briefly summarized in Theorem 1 in the Supplement, and apply them in the sampling step in Algorithm 1. Denoting the target density as $p(z; \lambda)$ with given $\lambda$, Theorem 1 shows that SGLD/SGHMC, by utilizing $\frac{\partial}{\partial z} \log p(z; \lambda)$, yields a non-homogeneous Markov chain $\left\{ z^{(l)}, l \geq 1 \right\}$, which converges to the equilibrium distribution $p(z; \lambda)$.

By letting $z \triangleq (x, h)$, $p(z; \lambda) \triangleq p_\theta(x) q_\phi(h|x)$, $\lambda \triangleq (\theta, \phi)^T$ in Theorem 1, we can perform the sampling step in Algorithm 1 by running $|\mathcal{M}|$ parallel chains, each chain being executed as shown

---

[4]Note that during training, $\sigma^2$ is absorbed into the learning rates and does not need to be estimated.
[5]Such optimization using two objectives is employed in a number of familiar learning methods, such as GAN with $\log D$ trick (Goodfellow et al., 2014), wake-sleep algorithm (Hinton et al., 1995).

---

**Algorithm 2** Sampling from $p_\theta(x)q_\phi(h|x)$

---

1. Do ancestral sampling by the generator, namely first drawing $h' \sim p(h')$, and then drawing $x' \sim q_\phi(x'|h')$;
2. Starting from $(x', h') = z^{(0)}$, run finite steps of SGLD/SGHMC ($l = 1, \cdots, L$) to obtain $(x, h) = z^{(L)}$, which we call *sample revision*, according to Eq. (11)/(12).
**Return** $(x, h)$.

---

in Algorithm 2. In sample revision, the calculation of the gradient w.r.t. $h$, $\frac{\partial}{\partial h} \log p(z; \lambda) = \frac{\partial}{\partial h} \log q_\phi(h|x) = \frac{\partial}{\partial h} \log q_\phi(h, x)$, is straightforward. For the gradient w.r.t. $x$, we have

$$\frac{\partial}{\partial x} \log p(z; \lambda) = \frac{\partial}{\partial x} \log p_\theta(x) + \frac{\partial}{\partial x} \log q_\phi(h, x) - \frac{\partial}{\partial x} \log q_\phi(x) \approx \frac{\partial}{\partial x} \log p_\theta(x). \quad (6)$$

The reason is that $\frac{\partial}{\partial x} \log q_\phi(x)$ can be approximated by an unbiased estimate, as proved in Proposition 2 in the Supplement:

$$\frac{\partial}{\partial x} \log q_\phi(x) \approx \frac{\partial}{\partial x} \log q_\phi(h, x).$$

Therefore, we can use $\frac{\partial}{\partial x} \log p_\theta(x)$[6] as an unbiased estimate of the gradient $\frac{\partial}{\partial x} \log p(z; \lambda)$, and we can apply Theorem 1 in the Supplement with tractable gradients w.r.t. both $x$ and $h$.

**Remarks.** Intuitively, the generator gives a proposal $(x', h')$, and then the system follows the gradients of $p_\theta(x)$ and $q_\phi(h, x)$ (w.r.t. $x$ and $h$ respectively) to revise $(x', h')$ to $(x, h)$. The gradient terms pull samples moving to low energy region of the random field and adjust the latent code of the generator, while the noise term brings randomness. In this manner, we obtain Markov chain samples from $p_\theta(x)q_\phi(h|x)$. Note that finite steps in sample revision will produce biased estimates of the gradients $\nabla_\theta$ and $\nabla_\phi$ in Eq. (5). We did not find this to pose problems to the SGD optimization in practice, as similarly found in Bornschein & Bengio (2015) and Kuleshov & Ermon (2017), which work with biased gradient estimators.

### 2.3 SEMI-SUPERVISED LEARNING WITH INCLUSIVE NRFs

In the following, we apply our inclusive-NRF approach in the SSL setting to show its flexibility. Note that different models are needed in unsupervised and semi-supervised learning, because SSL needs to additionally consider labels apart from observations.

**Model definition.** In semi-supervised tasks, we consider the following RF for joint modeling of observation $x \in \mathbb{R}^{d_x}$ and class label $y \in \{1, \cdots, K\}$:

$$p_\theta(x, y) = \frac{1}{Z(\theta)} \exp\left[u_\theta(x, y)\right] \quad (7)$$

which is different from Eq.1 for unsupervised learning without labels. To implement the potential function $u_\theta(x, y)$, we consider a neural network $\Phi_\theta(x) : \mathbb{R}^{d_x} \to \mathbb{R}^K$, with $x$ as the input and the output size being equal to the number of class labels, $K$. Then we define $u_\theta(x, y) = onehot(y)^T \Phi_\theta(x)$, where $onehot(y)$ represents the one-hot encoding vector for the label $y$. In this manner, the conditional density $p_\theta(y|x)$ is the classifier, defined as follows:

$$p_\theta(y|x) = \frac{p_\theta(x, y)}{p_\theta(x)} = \frac{\exp\left[u_\theta(x, y)\right]}{\sum_y \exp\left[u_\theta(x, y)\right]} \quad (8)$$

which acts like multi-class logistic regression using $K$ logits calculated from $x$ by the neural network $\Phi_\theta(x)$. And we do not need to calculate $Z(\theta)$ for classification. The auxiliary generator is implemented the same as in Eq. 3, i.e. an unconditional generator.

With the definition the joint density in Eq. 7, it can be shown that, with abuse of notation, the marginal density $p_\theta(x) = \frac{1}{Z(\theta)} \exp\left[u_\theta(x)\right]$ where $u_\theta(x) \triangleq log \sum_y \exp\left[u_\theta(x, y)\right]$.

---

[6]Note that $\frac{\partial}{\partial x} \log p_\theta(x) = \frac{\partial}{\partial x} u_\theta(x)$ does not require the calculation of the normalizing constant.

**Model learning.** Suppose that among the data $\mathcal{D} = \{\tilde{x}_1, \cdots, \tilde{x}_n\}$, only a small subset of the observations, for example the first $m$ observations, have class labels, $m \ll n$. Denote these labeled data as $\mathcal{L} = \{(\tilde{x}_1, \tilde{y}_1), \cdots, (\tilde{x}_m, \tilde{y}_m)\}$. Then we can formulate the semi-supervised learning as jointly optimizing

$$\begin{cases} \min_{\theta} KL\left[\tilde{p}(\tilde{x})||p_\theta(\tilde{x})\right] - \alpha_d \sum_{(\tilde{x},\tilde{y})\sim\mathcal{L}} log p_\theta(\tilde{y}|\tilde{x}) \\ \min_{\phi} KL\left[p_\theta(x)||q_\phi(x)\right] \end{cases} \tag{9}$$

which are defined by hybrids of generative and discriminative criteria, similar to Zhu (2006); Larochelle et al. (2012); Kingma et al. (2014). The hyper-parameter $\alpha_d$ controls the relative weight between generative and discriminative criteria. Similar to deriving Eq. (5), it can be easily seen that the gradients w.r.t. $\theta$ and $\phi$ (to be ascended) are defined as follows:

$$\begin{cases} \nabla_\theta^{\text{semi}} = E_{\tilde{p}(\tilde{x})}\left[\nabla_\theta log p_\theta(\tilde{x})\right] + \alpha_d \sum_{(\tilde{x},\tilde{y})\sim\mathcal{L}} \nabla_\theta log p_\theta(\tilde{y}|\tilde{x}) \\ \quad = E_{\tilde{p}(\tilde{x})}\left[\nabla_\theta u_\theta(\tilde{x})\right] - E_{p_\theta(x)}\left[\nabla_\theta u_\theta(x)\right] + \alpha_d \sum_{(\tilde{x},\tilde{y})\sim\mathcal{L}} \nabla_\theta log p_\theta(\tilde{y}|\tilde{x}) \\ \nabla_\phi^{\text{semi}} = E_{p_\theta(x)}\left[\nabla_\phi log q_\phi(x)\right] = E_{p_\theta(x)q_\phi(h|x)}\left[\nabla_\phi log q_\phi(x,h)\right] \end{cases} \tag{10}$$

In practice, we calculate noisy gradient estimators, and apply minibatch based stochastic gradient descent (SGD) to solve the optimization problem Eq. (9), as shown in Algorithm 3 in the Supplement. Apart from the basic losses as shown in Eq. (9), there are some regularization losses that are found to be helpful to guide SSL learning and are presented in the Supplement. To conclude, we show that the inclusive-NRF can be easily applied to SSL. To the best of our knowledge, there are no priori studies in applying random fields to SSL. **The semi-supervised inclusive-NRF model defined above is novel itself for SSL.**

## 3 RELATED WORK

Comparison and connection of our inclusive-NRF approach with existing studies are provided in the following from three perspectives.

**Learning NRFs with auxiliary generators.** These studies are most relevant to this work, which aims to learn NRFs. The classic method for learning RFs is the SML method (Younes, 1989), which works with the single target model $p_\theta$. Compared to learning traditional RFs which mainly use linear potential functions, learning NRFs which use NN based nonlinear potential functions, is more challenging. A recent progress in learning NRFs as studied in Kim & Bengio (2016); Xie et al. (2016); Wang & Ou (2017); Kuleshov & Ermon (2017) is to jointly train the target random field $p_\theta(x)$ and an auxiliary generator $q_\phi(x)$. Different studies differ in the objective functions used in the joint training, and thus have different computational and statistical properties.

- It is shown in Proposition 3 in the Supplement that learning in Kim & Bengio (2016) minimizes the exclusive-divergence $KL[q_\phi||p_\theta]$ w.r.t. $\phi$, which involves the intractable entropy term and tends to enforce the generator to seek modes, yielding missing modes. We refer to this approach as exclusive-NRF.

- Learning in Wang & Ou (2017) and in this paper minimizes the inclusive-divergence $KL[p_\theta||q_\phi]$ w.r.t. $\phi$. But noticeably, this paper presents our innovation in development of NRFs for continuous data, which is fundamentally different from Wang & Ou (2017) for discrete data. The target NRF model, the generator and the sampler are all different. Wang & Ou (2017) mainly studies random field language models, using LSTM generators (autoregressive with no latent variables) and employing Metropolis independence sampler (MIS) - applicable for discrete data (natural sentences). In this paper, we mainly develop random field models for continuous data (images), using latent-variable generators and utilizing SGLD/SGHMC (with solid theoretical examination) to exploit gradient information in the continuous space.

- In Xie et al. (2016), motivated by interweaving maximum likelihood training of the random field $p_\theta$ and the latent-variable generator $q_\phi$, a joint training method is introduced.

Operationally, in learning $\theta$ and $\phi$, this method also uses Langevin sampling to generate samples. Two Langevin sampling steps are intuitively interleaved according to $\frac{\partial}{\partial x} \log p_\theta(x)$ and $\frac{\partial}{\partial h} \log q_\phi(h, x)$ separately. This is different from our sampling step, which moves $(h, x)$ jointly, as theoretically justified in section 2.2. Let $r(h, x)$ denote the distribution obtained by running the interleaved Langevin transitions starting from $(h, x) \sim q_\phi(h, x)$. Interpretation presented in Xie et al. (2016) relates their method to the following joint optimization problem:

$$\begin{cases} \min_\theta \{ KL\left[\tilde{p}(\tilde{x}) || p_\theta(\tilde{x})\right] - KL\left[r(h, x) || p_\theta(x)\right] \} \\ \min_\phi KL\left[r(h, x) || q_\phi(h, x)\right] \end{cases}$$

which is also different from ours as shown in Eq. (4). Thus, learning in Xie et al. (2016) does not aim to minimize the inclusive-divergence $KL[p_\theta || q_\phi]$ w.r.t. $\phi$.

- Learning in Kuleshov & Ermon (2017) minimizes the $\chi^2$-divergence $\chi^2[q_\phi || p_\theta] \triangleq \int \frac{(p_\theta - q_\phi)^2}{q_\phi}$ w.r.t. $\phi$, which also tends to drive the generator to cover modes. But this approach is severely limited by the high variance of the gradient estimator w.r.t. $\phi$, and is only tested on the simpler MNIST and Omniglot.

Additionally, different NRF studies also differ in models used in the joint training. For example, the target NRF used in this work is different from those in previous studies Kim & Bengio (2016); Wang & Ou (2017); Xie et al. (2016). The differences are: Kim & Bengio (2016) includes linear and squared terms in $u_\theta(x)$, Wang & Ou (2017) defines over sequences, and Xie et al. (2016) defines in the form of exponential tilting of a reference distribution (Gaussian white noise). There exist different choices for the generator, such as GAN models in Kim & Bengio (2016), LSTMs in Wang & Ou (2017), or latent-variable models in both Xie et al. (2016) and this work. All are easy to do sampling.

Moreover, all the previous NRF studies examine unsupervised learning, and none shows application or extension of their methods or models for semi-supervised learning.

**Monte Carlo sampling.** One step in our inclusive-NRF approach is to apply SGLD/SGHMC to draw samples from the target density $p_\theta$, starting from the proposal sample from the generator. Theoretically, improvements in NRF sampling methods could be potentially integrated into NRF learning algorithms. For example, it is recently studied in Levy et al. (2018) to learn MCMC transition kernels, also parameterized by neural networks, to improve the HMC sampling from the given target distribution. Integration into learning NRFs is interesting but outside the scope of this paper.

**Comparison and connection with GANs.** On the one hand, there are some efforts that aim to address the inability of GANs to provide sensible energy estimates for samples. The energy-based GANs (Zhao et al., 2017) proposes to view the discriminator as an energy function by designing an auto-encoder discriminator. The recent work in Dai et al. (2017a) connects Zhao et al. (2017) and Kim & Bengio (2016), and show another two approximations for the entropy term. However, it is known that as the generator converges to the true data distribution, the GAN discriminator converges to a degenerate uniform solution. This basically afflicts the GAN discriminator to provide density information, though there are some modifications. In contrast, our inclusive-NRFs, unlike GANs, naturally provide (unnormalized) density estimate. Moreover, none of the above energy-related GAN studies examine their methods or models for SSL, except in EBGAN which performs moderately.

On the other hand, there are interesting connections between inclusive-NRFs and GANs, as elaborated in section 11 in the Supplement. When interpreting the potential function $u_\theta(x)$ as the critic in Wasserstein GANs, inclusive-NRFs seem to be similar to Wasserstein GANs. A difference is that in optimizing $\theta$ in inclusive-NRFs, the generated samples are further revised by taking finite-step-gradient of $u_\theta(x)$ w.r.t. $x$. However, the critic in Wasserstein GANs can hardly be interpreted as an unnormalized log-density. Thus strictly speaking, inclusive-NRFs are not GAN-like.

## 4 EXPERIMENTS

We conduct a series of experiments to evaluate the performances of our approach (inclusive-NRFs) and various existing methods on synthetic and real-world datasets for both unsupervised and semi-supervised learning tasks, with both visual and numerical evaluation. We refer to the Supplement for experimental details and additional results.

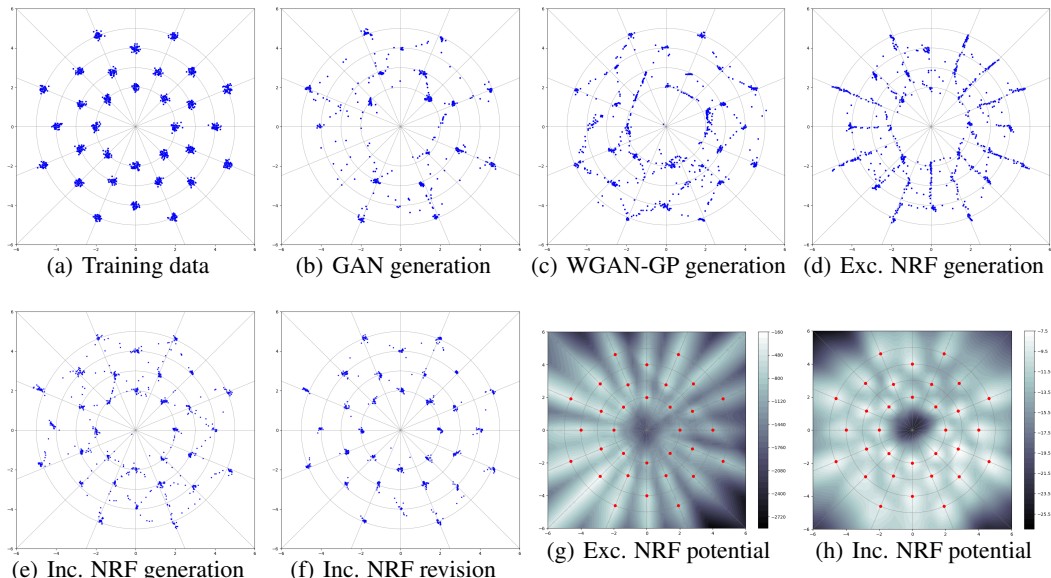

(a) Training data    (b) GAN generation    (c) WGAN-GP generation    (d) Exc. NRF generation

(e) Inc. NRF generation    (f) Inc. NRF revision    (g) Exc. NRF potential    (h) Inc. NRF potential

Figure 1: Comparison of different methods over GMM synthetic data. Stochastic generations from GAN with logD trick, WGAN-GP, Exclusive-NRF, **Inclusive-NRF generation** (i.e. sampling from the auxiliary generator) and **Inclusive-NRF revision** (i.e. after performing sample revision over samples from the auxiliary generator), are shown in (b)-(f) respectively. Inclusive-NRF generation and inclusive-NRF revision are two manners to generate samples, given a trained NRF. For both manners, the NRF model is trained with the sample revision step. Each generation contains 1,000 samples. The learned potentials $u_\theta(x)$ from exclusive and inclusive NRFs are shown in (g) and (h) respectively, where the red dots indicate the mean of each Gaussian component. Inclusive NRFs are clearly superior in learning data density and sample generation.

## 4.1 GMM SYNTHETIC EXPERIMENT

The synthetic data consist of 1,600 training examples generated from a 2D Gaussian mixture model (GMM) with 32 equally-weighted, low-variance ($\sigma = 0.1$) Gaussian components, uniformly laid out on four concentric circles as in Figure 1(a). The data distribution exhibits many modes separated by large low-probability regions, which makes it suitable to examine how well different learning methods can deal with multiple modes. For comparison, we experiment with GAN with logD trick (Goodfellow et al., 2014) and WGAN-GP (Gulrajani et al., 2017) for directed generative model, exclusive-NRF (Kim & Bengio, 2016) and inclusive-NRF for undirected generative model.

Figure 1 visually shows the generated samples from the trained models using different methods. Table 1 reports the "covered modes" and "realistic ratio" as numerical measures of how the multi-modal data are fitted, similarly as in Dumoulin et al. (2017). The main observations are as follows. (1) GAN suffers from mode missing, generating realistic but not diverse samples. WGAN-GP increases "covered modes" but decreases "realistic ratio". Inclusive-NRF performs much better than both GAN and WGAN-GP in sample generation. (2) Inclusive-NRF outperforms exclusive-NRF in both sample generation and density estimation. (3) After revision, samples from inclusive-NRF become more like real samples, achieving the best in both "covered modes" and "realistic ratio" metrics.

## 4.2 IMAGE GENERATION ON CIFAR-10

We examine both unsupervised and supervised learning over the widely used real-world dataset CIFAR-10 Krizhevsky (2009) for image generation. To evaluate generation quality quantitatively, we use inception score (IS) Salimans et al. (2016), and Frechet inception distance (FID) Heusel et al. (2017). Table 2 reports the inception score and FID for state of the art methods, for both unsupervised and supervised settings. The supervised learning of inclusive-NRF is conducted as a special case of semi-supervised learning over all labeled images ($m = n$), which uses unconditional generation. We use ResNet in this experiment, see section 12.2 in the Supplement for experimental details.

Table 1: Numerical evaluations over the GMM (32 components) synthetic data. The "**covered modes**" metric is defined as the number of covered modes by a set of generated samples. The "**realistic ratio**" metric is defined as the proportion of generated samples which are close to a mode. The measurement details are presented in section 12.1 in the Supplement. Mean and SD are from 10 independent runs.

| Methods | covered modes | realistic ratio |
|---|---|---|
| GAN with logD trick | $22.25 \pm 1.54$ | $0.90 \pm 0.01$ |
| WGAN-GP (Gulrajani et al., 2017) | $27.81 \pm 1.40$ | $0.74 \pm 0.04$ |
| Exclusive-NRF (Kim & Bengio, 2016) | $28.14 \pm 0.68$ | $0.73 \pm 0.03$ |
| **Inclusive-NRF generation** | $29.52 \pm 0.54$ | $0.84 \pm 0.01$ |
| **Inclusive-NRF revision** | $30.75 \pm 0.43$ | $0.97 \pm 0.01$ |

Table 2: Inception score (IS) and FID on CIFAR-10 for unsupervised and supervised learning.

| Methods | Unsupervised | | Supervised | |
|---|---|---|---|---|
| | IS | FID | IS | FID |
| DCGAN (Radford et al., 2015) | $6.16 \pm 0.07$ | | $6.58$ | |
| Improved-GAN (Salimans et al., 2016) | | | $8.09 \pm 0.07$ | |
| WGAN-GP (Gulrajani et al., 2017) | $7.86 \pm 0.07$ | | $8.42 \pm 0.10$ | |
| SGAN (Huang et al., 2017) | | | $8.59 \pm 0.12$ | |
| DFM (Warde-Farley & Bengio, 2017) | $7.72 \pm 0.13$ | | | |
| CT-GAN (Wei et al., 2018) | $8.12 \pm 0.12$ | | $8.81 \pm 0.13$ | |
| Fisher-GAN (Mroueh & Sercu, 2017) | $7.90 \pm 0.05$ | | $8.16 \pm 0.12$ | |
| BWGAN (Adler & Lunz, 2018) | $8.26 \pm 0.07$ | | | |
| SNGAN (Miyato et al., 2018) | $8.22 \pm 0.05$ | $21.7 \pm 0.21$ | | |
| **Inclusive-NRF generation** | $8.28 \pm 0.09$ | $20.9 \pm 0.25$ | $9.06 \pm 0.10$ | $18.1 \pm 0.23$ |

From the comparison results in Table 2, it can be seen that the proposed inclusive-NRF model achieves the best inception score over CIFAR-10, to the best of our knowledge, in both unsupervised and supervised settings. Some generated samples are shown in Figure 5(c)(d) for unsupervised and supervised settings respectively. We also show in the Supplement the capability of inclusive-NRFs in latent space interpolation (section 14) and conditional generation (section 15).

### 4.3 SEMI-SUPERVISED LEARNING RESULTS

For semi-supervised learning, we consider the three widely used benchmark datasets, namely MNIST (LeCun et al., 1998), SVHN (Netzer et al., 2011), and CIFAR-10 (Krizhevsky, 2009). As in previous work, we randomly sample 100, 1,000, and 4,000 labeled samples from MNIST, SVHN, and CIFAR-10 respectively during training, and use the standard data split for testing. See section 12.3 in the Supplement for experimental details. We also provide a SSL toy experiment in section 13 in the Supplement to help understanding how semi-supervised inclusive-NRF works.

It can be seen from Table 3 that semi-supervised inclusive-NRFs produce strong classification results on par with state-of-art DGM-based SSL methods. See Figure 5(a)(b) in the Supplement for generated samples. Bad-GANs achieve better classification results, but as indicated by the low inception score, their generation is much worse than semi-NRF-IAGs. In fact, among DGM-based SSL methods, inclusive-NRFs achieve the best performance in sample generation. **This is in contrast to the conflict of good classification and good generation, as observed in GAN-based SSL** (Salimans et al., 2016; Dai et al., 2017b). It is analyzed in Dai et al. (2017b) that good GAN-based SSL requires a bad generator[7]. This is embarrassing and in fact obviates the original idea of generative SSL - successful generative training, which indicates good generation, provides regularization for finding good classifiers (Zhu, 2006; Larochelle et al., 2012). In this sense, Bad-GANs could hardly be classified as a generative SSL method.

---

[7]This analysis is based on using the $(K + 1)$-class GAN-like discriminator objective for SSL. To the best of our knowledge, the conflict does not seem to be reported in previous generative SSL methods Zhu (2006); Larochelle et al. (2012) which use the $K$-class classifier like in semi-supervised inclusive-NRFs.

Table 3: Comparison with state-of-the-art methods on three benchmark datasets. "CIFAR-10 IS" means the inception score for samples generated by SSL models trained on CIFAR-10. "†" is obtained by running the released code accompanied by the corresponding papers. "-" means the results are not reported in the original work and without released code. "/" means not applicable, e.g. the models cannot generate samples stochastically. "‡" uses image data augmentation which significantly helps classification performance. The upper/lower blocks show generative/discriminative SSL methods respectively.

| Methods | error (%) MNIST | error (%) SVHN | error (%) CIFAR-10 | IS CIFAR-10 |
|---|---|---|---|---|
| CatGAN (Springenberg, 2016) | $1.91 \pm 0.10$ | - | $19.58 \pm 0.46$ | $3.57 \pm 0.13^{\dagger}$ |
| SDGM (Maaloe et al., 2016) | $1.32 \pm 0.07$ | $16.61 \pm 0.24$ | - | - |
| Ladder network (Rasmus et al., 2015) | $1.06 \pm 0.37$ | - | $20.40 \pm 0.47$ | / |
| ADGM (Maaloe et al., 2016) | $0.96 \pm 0.02$ | 22.86 | - | - |
| Improved-GAN (Salimans et al., 2016) | $0.93 \pm 0.07$ | $8.11 \pm 1.3$ | $18.63 \pm 2.32$ | $3.87 \pm 0.03$ |
| EBGAN (Zhao et al., 2017) | $1.04 \pm 0.12$ | - | - | - |
| ALI (Dumoulin et al., 2017) | - | $7.42 \pm 0.65$ | $17.99 \pm 1.62$ | - |
| Triple-GAN (Li et al., 2017) | $0.91 \pm 0.58$ | $5.77 \pm 0.17$ | $16.99 \pm 0.36$ | $5.08 \pm 0.09$ |
| Triangle-GAN (Gan et al., 2017) | - | - | $16.80 \pm 0.42$ | - |
| BadGAN (Dai et al., 2017b) | $0.80 \pm 0.10$ | $4.25 \pm 0.03$ | $14.41 \pm 0.30$ | $3.46 \pm 0.11^{\dagger}$ |
| Sobolev-GAN (Mroueh et al., 2018) | - | - | $15.77 \pm 0.19$ | - |
| **Semi-supervised inclusive-NRF** | $0.97 \pm 0.10$ | $5.84 \pm 0.15$ | $15.12 \pm 0.36$ | $7.72 \pm 0.09$ |
| Results below this line cannot be directly compared to those above. | | | | |
| VAT small (Miyato et al., 2017) | 1.36 | 6.83 | 14.87 | / |
| Π model‡ (Laine & Aila, 2017) | - | $4.82 \pm 0.17$ | $12.36 \pm 0.31$ | / |
| Temporal Ensembling‡ (Laine & Aila, 2017) | - | $4.42 \pm 0.16$ | $12.16 \pm 0.31$ | / |
| Mean Teacher‡ (Tarvainen & Valpola, 2017) | - | $3.95 \pm 0.19$ | $12.31 \pm 0.28$ | / |
| VAT+EntMin‡ (Miyato et al., 2017) | - | 3.86 | 10.55 | / |
| CT-GAN‡ (Wei et al., 2018) | $0.89 \pm 0.13$ | - | $9.98 \pm 0.21$ | / |

Finally, note that some discriminative SSL methods, as listed in the lower block in Table 3 also produce superior performances, by utilizing data augmentation and consistency regularization. However, these methods are unable to generate (realistic) samples. It can be seen that discriminative SSL methods utilize different regularization from generative SSL methods and cannot be directly compared to generative SSL methods. Their combination, as an interesting future work, could yield further performance improvement.

## 4.4 ABLATION STUDY

We report the results of ablation study of our inclusive-NRF method on CIFAR-10 in Table 4. In this experiment, we use the standard CNN (Miyato et al., 2018) for unsupervised learning and the same networks as those used in Table 3 for semi-supervised learning. See section 12.4 in the Supplement for experimental details. We analyze the effects of different settings in model training, such as using SGLD or SGHMC and the revision step $L = 1/5/10$ used. For each training setting, we also compare the two manners to generate samples - whether applying sample revision or not in inference (generating samples) given a trained NRF, as previously illustrated in Figure 1 over synthetic GMM data. The main observations are as follows.

First, given a trained NRF, after revision (i.e. following the gradient of the RF's potential $u_\theta(x)$ w.r.t. $x$), the quality (IS) of samples is always improved, as shown by the consistent IS improvement from the second column (generation) to the third (revision). This is in accordance with the results in the GMM synthetic experiments. Moreover, noting that in revision, it is the the estimated density $p_\theta$ that guides the samples towards low energy region of the random field. **This demonstrates one benefit of random field modeling, which, unlike GANs, can learn density estimate about the data manifold.**

Second, a row-wise reading of Table 4 reveals that with more revision steps and using SGHMC in training, the SSL classification performance is improved. Utilizing SGHMC in inclusive-NRFs to

Table 4: Ablation study of our inclusive-NRF method on CIFAR-10, regarding the effects of using SGLD or SGHMC in training and of applying sample revision in inference (generating samples). Mean and SD are from 5 independent runs for each training setting. In each training setting, for unsupervised learning, two manners to generate samples given a trained NRF are compared, as previously illustrated in Figure 1 over synthetic GMM data. We examine generated samples (i.e. directly from the generator) and revised samples (i.e. after sample revision) respectively, in term of inception scores (IS). For semi-supervised learning, we examine the classification error rates.

| Training | Unsupervised | | Semi-supervised |
|----------|--------------|--------------|-----------------|
| Setting | Generation IS | Revision IS | error (%) |
| SGLD $L = 1$ | $7.47 \pm 0.15$ | $7.53 \pm 0.13$ | $17.08 \pm 0.39$ |
| SGLD $L = 5$ | $7.44 \pm 0.16$ | $7.49 \pm 0.12$ | $16.15 \pm 0.44$ |
| SGLD $L = 10$ | $7.43 \pm 0.18$ | $7.50 \pm 0.13$ | $15.60 \pm 0.31$ |
| SGHMC $L = 10$ | $7.46 \pm 0.12$ | $7.57 \pm 0.10$ | $15.12 \pm 0.36$ |

exploit gradient information with momentum yields better performance than simple SGLD as used in Xie et al. (2016). It is also found that more revision steps in model training do not significantly improve unsupervised IS. So we can use $L = 1$ in unsupervised learning for generation, which can reduce the computational cost.

## 5 DISCUSSION AND CONCLUSION

In this paper we develop the inclusive-NRF approach, which learns NRFs with inclusive auxiliary generators and particularly for continuous data, exploits gradient information in model sampling with solid theoretical examination. Extensive empirical evaluations show that inclusive-NRFs obtain state-of-the-art sample generation quality and achieve strong semi-supervised learning results on par with state-of-the-art DGMs. The superior performances presumably are attributed to **the two distinctive features in inclusive-NRFs - introducing the inclusive-divergence minimized auxiliary generator and utilizing sample revision by SGLD/SGHMC**. Intuitively, the revised samples from the RF will guide the training of the generator, and subsequently the generator will propose samples for the RF to sense the data manifold. This forms positive interactions between the random field and the generator, which enables successful joint training of both models.

The new approach enables us to flexibly use NRFs in unsupervised, supervised and semi-supervised settings and successfully train them in a black-box manner. Interesting future work will consider inclusive-NRFs in more challenging tasks, e.g. unsupervised and semi-supervised learning with sequential data (e.g. speech, language, video, etc.).

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

# Supplement for "Learning Neural Random Fields with Inclusive Auxiliary Generators"

## 6 PROOF OF PROPOSITION 1

**Proposition 1.** *Both lines of Eq.(5) for gradient calculations hold.*

*Proof.* The first line of Eq.(5) can be obtained by directly taking derivative of $KL\left[\tilde{p}(\tilde{x})||p_\theta(\tilde{x})\right]$ w.r.t. $\theta$, as shown below,

$$\frac{\partial}{\partial \theta} KL\left[\tilde{p}(\tilde{x})||p_\theta(\tilde{x})\right] = \frac{\partial}{\partial \theta} \int \tilde{p}(\tilde{x}) \log \frac{\tilde{p}(\tilde{x})}{p_\theta(\tilde{x})} d\tilde{x} = -\int \tilde{p}(\tilde{x}) \frac{\partial}{\partial \theta} \log p_\theta(\tilde{x}) d\tilde{x},$$

and then applying the basic formula of Eq. (2).

For the second line, by direct calculation, we first have

$$E_{q_\phi(h|x)}\left[\nabla_\phi \log q_\phi(h|x)\right] = \int q_\phi(h|x) q_\phi(h|x)^{-1} \nabla_\phi q_\phi(h|x) dh$$

$$= \int \nabla_\phi q_\phi(h|x) dh = \nabla_\phi \int q_\phi(h|x) dh = \nabla_\phi 1 = 0.$$

Then combining $\frac{\partial}{\partial \phi} KL\left[p_\theta(x)||q_\phi(x)\right] = -E_{p_\theta(x)}\left[\nabla_\phi \log q_\phi(x)\right]$ and

$$\nabla_\phi \log q_\phi(x) = E_{q_\phi(h|x)}\left[\nabla_\phi \log q_\phi(x)\right] = E_{q_\phi(h|x)}\left[\nabla_\phi \log q_\phi(x,h) - \nabla_\phi \log q_\phi(h|x)\right]$$

$$= E_{q_\phi(h|x)}\left[\nabla_\phi \log q_\phi(x,h)\right].$$

will give the second line of Eq.(5). □

## 7 SGLD/SGHMC

**Theorem 1.** *Denote the target density as $p(z; \lambda)$ with given $\lambda$. Assume that one can compute a noisy, unbiased estimate $\Delta(z, \xi; \lambda)$ to the gradient $\frac{\partial}{\partial z} \log p(z; \lambda)$, where $\xi$ is an auxiliary random variable which contains all the randomness involved in constructing the estimate, namely $E\left[\Delta(z, \xi; \lambda)\right] = \frac{\partial}{\partial z} \log p(z; \lambda)$. Assume the stability Assumptions 4 in Teh et al. (2016) holds.*

*For a sequence of asymptotically vanishing time-steps $\{\delta_l, l \geq 1\}$ (satisfying $\sum_{l=1}^{\infty} \delta_l = \infty$ and $\sum_{l=1}^{\infty} \delta_l^2 < \infty$), an i.i.d. sequence $\eta^{(l)}$, and an independent and i.i.d. sequence $\xi_l$ of auxiliary random variables, $l \geq 1$, the SGLD iterates as follows, starting from $z^{(0)}$:*

$$z^{(l)} = z^{(l-1)} + \frac{\delta_l}{2} \Delta(z^{(l-1)}, \xi_l; \lambda) + \sqrt{\delta_l} \eta^{(l)}, \eta^{(l)} \sim \mathcal{N}(0, I), l = 1, \cdots \qquad (11)$$

*Starting from $z^{(0)}$ and $v^{(0)} = 0$, the SGHMC iterates as follows:*

$$\begin{cases} v^{(l)} = \beta v^{(l-1)} + \frac{\delta_l}{2} \Delta(z^{(l-1)}, \xi_l; \lambda) + \sqrt{\delta_l} \eta^{(l)}, \eta^{(l)} \sim \mathcal{N}(0, I) \\ z^{(l)} = z^{(l-1)} + v^{(l)}, l = 1, \cdots \end{cases} \qquad (12)$$

*Then in both cases, the non-homogeneous Markov chain $\{z^{(l)}, l \geq 1\}$ converges to the equilibrium distribution $p(z; \lambda)$.*

## 8 PROOF OF PROPOSITION 2

**Proposition 2.** *Let* $z \triangleq (x, h)$, $p(z; \lambda) \triangleq p_\theta(x)q_\phi(h|x)$, $\lambda \triangleq (\theta, \phi)^T$ *in Theorem 1. The initial value* $(x^{(0)}, h^{(0)})$ *is obtained from ancestral sampling by the generator. The SGLD/SGHMC as shown in Eq.(11)/(12) iteratively generates* $(x^{(m)}, h^{(m)})$, $m = 1, \cdots$. *Then,* $\frac{\partial}{\partial x^{(m)}} \log q_\phi(h^{(m)}, x^{(m)})$ *is an unbiased estimate of the gradient* $\frac{\partial}{\partial x^{(m)}} \log q_\phi(x^{(m)})$, $m = 0, 1, \cdots$.

*Proof.* Note that Langevin dynamics and Hamiltonian dynamics are reversible Neal (2011). Thus the SGLD/SGHMC transitions Eq.(11)/(12) satisfy the detailed balance condition:

$$\pi(h^{(m-1)}, x^{(m-1)})K(h^{(m)}, x^{(m)}|h^{(m-1)}, x^{(m-1)}) = \pi(h^{(m)}, x^{(m)})K(h^{(m-1)}, x^{(m-1)}|h^{(m)}, x^{(m)}),$$

where $\pi(\cdot)$ denotes the target density, and $K(\cdot|\cdot)$ denotes the transition kernel. Also note that $E_{h\sim q_\phi(h|x)}\left[\frac{\partial}{\partial x}\log q_\phi(h, x)\right] = \frac{\partial}{\partial x}\log q_\phi(x)$. Thus if we show that $h^{(m)}$ is indeed drawn from $q_\phi(h^{(m)}|x^{(m)})$ during sample revision, $m = 0, 1, \cdots$, then the unbiasedness will hold.

Denote by $\pi^{(m)}(h^{(m)}, x^{(m)})$ the state-occupation density at step $m$. Then we need to show that $\pi^{(m)}(h^{(m)}|x^{(m)})$ actually follows $\pi(h^{(m)}|x^{(m)})$, i.e. $q_\phi(h^{(m)}|x^{(m)})$, $m = 0, 1, \cdots$.

First, it is obvious that this holds for $(h^{(0)}, x^{(0)})$. Then, we proceed by mathematical induction. Suppose $\pi^{(m-1)}(h^{(m-1)}|x^{(m-1)}) = \pi(h^{(m-1)}|x^{(m-1)})$. Then we have

$$\pi^{(m-1)}(h^{(m-1)}, x^{(m-1)})K(h^{(m)}, x^{(m)}|h^{(m-1)}, x^{(m-1)})$$
$$=\pi^{(m-1)}(h^{(m-1)}, x^{(m-1)})\frac{\pi(h^{(m)}, x^{(m)})}{\pi(h^{(m-1)}, x^{(m-1)})}K(h^{(m-1)}, x^{(m-1)}|h^{(m)}, x^{(m)}). \tag{13}$$

Integrating out $(h^{(m-1)}, x^{(m-1)})$ from both sides of Eq.13, we obtain

$$\pi^{(m)}(h^{(m)}, x^{(m)}) = \pi(h^{(m)}, x^{(m)}) \sum_{x^{(m-1)}} \frac{\pi^{(m-1)}(x^{(m-1)})}{\pi(x^{(m-1)})}K(x^{(m-1)}|h^{(m)}, x^{(m)})$$
$$= \pi(h^{(m)}, x^{(m)}) \sum_{x^{(m-1)}} \frac{\pi^{(m-1)}(x^{(m-1)})}{\pi(x^{(m-1)})}K(x^{(m-1)}|x^{(m)})$$

where the second equality, i.e. $K(x|h', x') = K(x|x')$, holds because in the SGLD/SGHMC transitions Eq.(11)/(12), generating next step $x$ only depends on current $x'$ and is independent of current $h'$. Then we have

$$\pi^{(m)}(h^{(m)}|x^{(m)}) = \pi(h^{(m)}|x^{(m)})\frac{\pi(x^{(m)})}{\pi^{(m)}(x^{(m)})} \sum_{x^{(m-1)}} \frac{\pi^{(m-1)}(x^{(m-1)})}{\pi(x^{(m-1)})}K(x^{(m-1)}|x^{(m)})$$
$$= \pi(h^{(m)}|x^{(m)})$$

where the second equality holds because we have

$$\frac{\pi(x^{(m)})}{\pi^{(m)}(x^{(m)})} \sum_{x^{(m-1)}} \frac{\pi^{(m-1)}(x^{(m-1)})}{\pi(x^{(m-1)})}K(x^{(m-1)}|x^{(m)})$$
$$=\frac{1}{\pi^{(m)}(x^{(m)})} \sum_{x^{(m-1)}} \pi^{(m-1)}(x^{(m-1)})\frac{K(x^{(m-1)}|x^{(m)})\pi(x^{(m)})}{\pi(x^{(m-1)})}$$
$$=1$$

Thereby, we show $h^{(m)} \sim \pi(h^{(m)}|x^{(m)})$, i.e. $q_\phi(h^{(m)}|x^{(m)})$. This concludes the inductive step. $\square$

## 9 SEMI-SUPERVISED LEARNING WITH INCLUSIVE-NRFs

Apart from the basic losses, as shown in Eq.10, in applying inclusive-NRFs in SSL, there are some regularization losses that are helpful to guide the semi-supervised learning.

---

**Algorithm 3** Semi-supervised learning of inclusive-NRFs

---
**repeat**
    Sampling:
    Draw a unsupervised minibatch $\mathcal{U} \sim \tilde{p}(\tilde{x})p_\theta(x)q_\phi(h|x)$ and a supervised minibatch $\mathcal{S} \sim \mathcal{L}$;
    Updating:
    Update $\theta$ by ascending:
    $\frac{1}{|\mathcal{U}|} \sum_{(\tilde{x},x,h)\sim\mathcal{U}} [\nabla_\theta u_\theta(\tilde{x}) - \nabla_\theta u_\theta(x)] + \alpha_d \frac{1}{|\mathcal{S}|} \sum_{(\tilde{x},\tilde{y})\sim\mathcal{S}} [\nabla_\theta log p_\theta(\tilde{y}|\tilde{x})]$
    $- \frac{1}{|\mathcal{U}|} \sum_{(\tilde{x},x,h)\sim\mathcal{U}} \left[ \alpha_c \nabla_\theta H(p_\theta(y|\tilde{x})) + \alpha_p \nabla_\theta [u_\theta(\tilde{x})]^2 \right]$;
    Update $\phi$ by ascending:
    $\frac{1}{|\mathcal{U}|} \sum_{(\tilde{x},x,h)\sim\mathcal{U}} \nabla_\phi log q_\phi(x,h)$;
**until** convergence

---

**Confidence loss.** Similar to Springenberg (2016); Li et al. (2017), we add the minimization of the conditional entropy of $p_\theta(y|\tilde{x})$ averaged over training data to the loss w.r.t. $\theta$ (i.e. the first line in Eq.9) as follows:

$$L_c(\theta) = E_{\tilde{p}(\tilde{x})} [H(p_\theta(y|\tilde{x}))] = -E_{\tilde{p}(\tilde{x})} \left[ \sum_y p_\theta(y|\tilde{x}) \log p_\theta(y|\tilde{x}) \right]$$

In this manner, we encourage the classifier $p_\theta(y|x)$ derived from the RF to make classifications confidently. In practice, we use stochastic gradients of $L_c(\theta)$ over minibatches in optimizing $\theta$, as shown in Algorithm 3.

**Potential control loss.** For random fields, the data log-likelihood $log p_\theta(\tilde{x})$ is determined relatively by the potential value $u_\theta(\tilde{x})$. To avoid the potential values not to increase unreasonably, we could control the squared potential values, by minimizing:

$$L_p(\theta) = E_{\tilde{p}(\tilde{x})} [u_\theta(\tilde{x})]^2$$

In this manner, the potential values would be attracted to zeros. In practice, we use stochastic gradients of $L_p(\theta)$ over minibatches in optimizing $\theta$, as shown in Algorithm 3.

## 10 PROOF OF PROPOSITION 3

**Proposition 3.** *For the RF as defined in Eq. 1, we have the following evidence upper bound:*

$$log p_\theta(\tilde{x}) = \mathcal{U}(\tilde{x};\theta,\phi) - KL(q_\phi(x)||p_\theta(x)) \leq \mathcal{U}(\tilde{x};\theta,\phi),$$
$$\mathcal{U}(\tilde{x};\theta,\phi) \triangleq u_\theta(\tilde{x}) - \left( E_{q_\phi(x)}[u_\theta(x)] + H[q_\phi(x)] \right).$$

*Proof.* Note that $log p_\theta(\tilde{x}) = u_\theta(\tilde{x}) - log Z(\theta)$. And we have the following lower bound on $Z(\theta)$
$log Z(\theta) = log \int \exp(u_\theta(x))dx = log \int q_\phi(x)\frac{\exp(u_\theta(x))}{q_\phi(x)}dx \geq \int q_\phi(x)log\frac{\exp(u_\theta(x))}{q_\phi(x)}dx$. This can be also seen from:

$$\int q_\phi(x)u_\theta(x)dx = \int q_\phi(x)log p_\theta(x)dx + log Z(\theta)$$

$$= -KL(q_\phi(x)||p_\theta(x)) + log Z(\theta) + \int q_\phi(x)log q_\phi(x)dx.$$

$\square$

Furthermore, it can be seen that learning in Kim & Bengio (2016) amounts to optimizing the following evidence upper bound:

$$\max_\theta \min_\phi \mathcal{U}(\tilde{x};\theta,\phi),$$

which is unfortunately not revealed in this manner in Kim & Bengio (2016).

## 11 CONNECTION BETWEEN INCLUSIVE-NRFS AND GANS

Note that for the generator as defined in Eq. 3, we have the following joint density

$$logq_\phi(x, h) = -\frac{1}{2\sigma^2}||x - g_\phi(h)||^2 + constant.$$

The generator parameter $\phi$ is updated according to Eq. 5, which is rewritten as follows:

$$E_{p_\theta(x)q_\phi(h|x)}\left[\nabla_\phi logq_\phi(x, h)\right] = 0$$

Specifically, we draw $(h', x') \sim q_\phi$ and then perform one-step SGLD to obtain $(h, x)$. To simply the analysis of the connection, suppose $h \approx h'$, $x' \approx g_\phi(h') \approx g_\phi(h)$. Then we have

$$
\begin{aligned}
x &= x' + \frac{\delta_1}{2}\left[\frac{\partial}{\partial x}logp_\theta(x)\right]\Bigg|_{x=x'} + \sqrt{\delta_1}\eta^{(1)}, \eta^{(1)} \sim \mathcal{N}(0, I) \\
x - g_\phi(h) &\approx \frac{\delta_1}{2}\left[\frac{\partial}{\partial x}logp_\theta(x)\right]\Bigg|_{x=g_\phi(h)} = \frac{\delta_1}{2}\left[\frac{\partial}{\partial x}u_\theta(x)\right]\Bigg|_{x=g_\phi(h)}
\end{aligned}
\tag{14}
$$

The gradient in the updating step in Algorithm 1 becomes:

$$
\begin{aligned}
\nabla_\phi logq_\phi(x, h) &= \frac{1}{\sigma^2}\left[\frac{\partial}{\partial\phi}g_\phi(h)\right][x - g_\phi(h)] \\
&\approx \frac{1}{\sigma^2}\left[\frac{\partial}{\partial\phi}g_\phi(h)\right]\frac{\delta_1}{2}\left[\frac{\partial}{\partial x}u_\theta(x)\right]\Bigg|_{x=g_\phi(h)} \\
&= \frac{1}{\sigma^2}\frac{\delta_1}{2}\left[\frac{\partial}{\partial\phi}u_\theta(g_\phi(h))\right]
\end{aligned}
$$

where $\frac{\partial}{\partial\phi}g_\phi(h)$ is a matrix of size $dim(\phi) \times dim(x)$. Therefore, the inclusive-NRF Algorithm 1 can be viewed to perform the following steps:

1. Draw an empirical example $\tilde{x} \sim p_0$.
2. Draw $h \sim p(h)$, $x' = g_\phi(h)$, and generate $x$ by one-step-gradient according to Eq. 14.
3. Update $\theta$ by ascending: $\nabla_\theta u_\theta(\tilde{x}) - \nabla_\theta u_\theta(x)$.
4. Update $\phi$ by descending: $-\frac{\partial}{\partial\phi}u_\theta(g_\phi(h))$.

Now suppose that we interpret the potential function $u_\theta(x)$ as the discriminator in GANs (or the critic in Wasserstein GANs), which assign high scalar scores to empirical samples $\tilde{x} \sim p_0$ and low scalar scores to generated samples $x$. Then, the inclusive-NRF training could be viewed as playing a two-player minimax game:

$$\min_\phi \max_\theta E_{\tilde{x}\sim p_0}\left[u_\theta(\tilde{x})\right] - E_{h\sim p(h)}\left[u_\theta(g_\phi(h))\right], \tag{15}$$

*except that* in optimizing $\theta$, the generated sample are further revised by taking one-step-gradient of $u_\theta(x)$ w.r.t. $x$ (as shown in the above Step 2). The discriminator $u_\theta$ is trained to discriminate between empirical samples and generated samples, while the generator $q_\phi$ is trained to fool the discriminator by assigning higher scores to generated samples. From the above analysis, we find some interesting connections between inclusive-NRFs and existing studies in GANs.

- The optimization shown in Eq. 15 is in fact the same as that in Wasserstein GANs (Theorem 3 in Arjovsky et al. (2017)), *except that* in Wasserstein GANs, the critic $u_\theta(x)$ is constrained to be 1-Lipschitz continuous. So hopefully we can improve the inclusive-NRF training by constraining the discriminator $u_\theta(x)$ to be 1-Lipschitz continuous, e.g. by utilizing the recently developed technique of spectral normalization of weight matrices in the discriminator as in Miyato et al. (2018).

- To optimize $\theta$, the generated sample is obtained by taking one-step-gradient of $u_\theta(x)$ w.r.t. $x$. The tiny perturbation guided by the gradient to increase the score for the generated sample in fact creates an adversarial example. A similar idea is presented in Liu & Hsieh (2018) that when feeding real samples to the discriminator, 5 steps of PGD (Projected Gradient Descent) attack is taken to decrease the score to create adversarial samples. It is shown in Liu & Hsieh (2018) that training the discriminator with adversarial examples significantly improves the GAN traning. Hopefully in training the discriminator in inclusive-NRFs, the adversarial attack could be increasing scores for generated samples, or decreasing scores for real samples, or a mixed one.

- The above analysis assume the use of one-step SGLD. It can be seen that running finite steps of SGLD in sample revision in fact create adversarial samples to fool the discriminator.

## 12 DETAILS OF EXPERIMENTS

### 12.1 GMM SYNTHETIC EXPERIMENT

In the GMM experiment, we use the following procedure to estimate the metrics "covered modes" and "realistic ratio" for each trained model.

1. Stochastically generate 100 samples.

2. A mode is defined to be covered (not missed) if there exist generated samples located closely to the mode (with squared distance $< 0.02$), and those samples are said to be realistic.

3. Count how many modes are covered and calculate the proportion of realistic samples.

4. Repeat the above steps 100 times and perform averaging.

For each method, we independently train 10 models and calculate the mean and standard deviation (SD) across the 10 independent runs.

The network architectures and hyperparameters are the same for all methods, as listed in Table 5. We use SGLD Welling & Teh (2011) for inclusive-NRFs on this synthetic dataset, with empirical revision hyperparameters $\delta_l = 0.01$.

### 12.2 IMAGE GENERATION ON CIFAR-10

**Network architectures.** For convenience, we refer to the two neural networks in implementing the potential $u_\theta$ and the generator $q_\phi$ in NRFs as the potential network and the generator network, respectively. For comparison of different methods, we use the same network architectures as in Table 4 in (Miyato et al., 2018) (ResNet using spectral normalization) for unsupervised learning of NRFs. For supervised learning, we use the semi-supervised inclusive-NRF Algorithm 3 over all labeled images. The difference in network architectures used for semi-supervised and unsupervised learning of inclusive-NRFs is that for SSL, the output layer of the potential network contains $K = 10$ scalar units, while a single scalar output unit is used for unsupervised learning.

**Hyperparameters.** We use Adam optimizer with the hyperparameter ($\beta_1 = 0, \beta_2 = 0.9$ and $\alpha = 0.0003$ for random fields, $\alpha = 0.0001$ for generators). For sample revision for inclusive-NRFs, we empirically choose SGLD with $L = 1$ ($\delta_l = 0.003$). More revision steps do not significantly improve unsupervised IS, as discussed in section 4.4 Note that we use the potential control loss in both unsupervised ($\alpha_p = 0.1$) and supervised ($\alpha_d = 1, \alpha_p = 0.1$) settings, which is found beneficial for stable training.

**Evaluation.** Figure 5(c)(d) show the generated samples from inclusive-NRFs for unsupervised and supervised settings respectively. We compute inception score (IS) and Frechet inception distance (FID) in the same way as in Miyato et al. (2018). We trained 10 models with different random seeds, and then generate 5000 images 10 times and compute the average inception score and the standard deviation. We compute FID between the true distribution and the generated distribution empirically over 10000 (test set) and 5000 samples.

## 12.3 SEMI-SUPERVISED EXPERIMENT ON MNIST, SVHN AND CIFAR-10

The network architectures (taken from the released code from Salimans et al. (2016) and widely used in Li et al. (2017); Dai et al. (2017b)) and hyperparameters for semi-supervised inclusive-NRFs on MNIST, SVHN and CIFAR-10 are listed in Table 6, Table 7 and Table 8 respectively. We use SGHMC for semi-supervised inclusive-NRFs for all three datasets, with empirical revision hyperparameters ($\beta = 0.5, \delta_l = 0.003$) for MNIST and CIFAR-10, and ($\beta = 0.5, \delta_l = 0.01$) for SVHN. The confidence loss is employed for semi-supervised inclusive-NRFs on MNIST and SVHN, and the potential control loss is employed on CIFAR-10.

Figure 5(a)(b) show the generated samples from semi-supervised inclusive-NRFs trained over SVHN and CIFAR-10 respectively.

## 12.4 ABLATION STUDY OF INCLUSIVE-NRFs ON CIFAR-10

For unsupervised learning, we use the same networks as in Table 3 in Miyato et al. (2018) (standard CNN using spectral normalization). We use Adam optimizer with the hyperparameter ($\alpha = 0.0002, \beta_1 = 0, \beta_2 = 0.9$). For semi-supervised learning, the experimental setting is the same as in section 12.3 including the networks, number of labels, etc. For different revision steps, we use ($\delta_l = 0.003$) for SGLD, and ($\beta = 0.5, \delta_l = 0.003$) for SGHMC. The potential control loss is employed in both unsupervised ($\alpha_p = 0.1$) and semi-supervised ($\alpha_d = 100, \alpha_p = 0.1$) learning.

## 13 SSL TOY EXPERIMENT

In Figure 2, we present the performance of semi-supervised inclusive-NRFs for SSL on a synthetic dataset, which emphasizes that inclusive-NRFs can provide (unnormalized) density estimates for $p_\theta(x)$, $p_\theta(x, y = 1)$ and $p_\theta(x, y = 2)$. In contrast, the use of GANs as general purpose probabilistic generative models has been limited by the difficulty in using them to provide density estimates or even unnormalized potential values for sample evaluation.

The dataset is a 2D GMM with 16 Gaussian components, uniformly laid out on two concentric circles. The two circles represent two different classes, each class with 4 labeled data and 400 unlabeled data. The network architectures are the same as in Table 5, except that the neural network which implement the potential function $u_\theta(x, y)$ for SSL now has two units in the output.

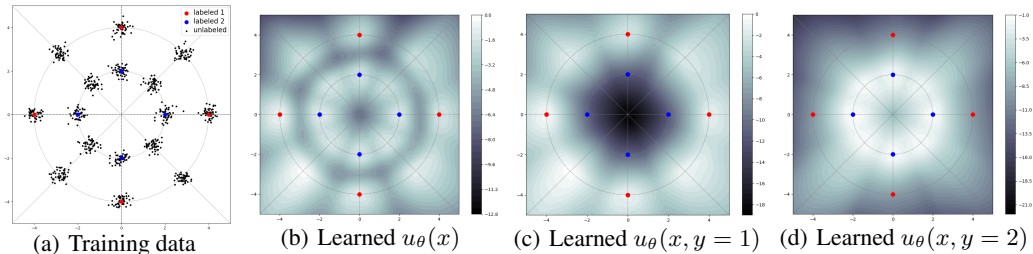

(a) Training data     (b) Learned $u_\theta(x)$     (c) Learned $u_\theta(x, y = 1)$     (d) Learned $u_\theta(x, y = 2)$

Figure 2: SSL toy experiment based on semi-supervised inclusive-NRFs. Each class has 4 labeled points, red dots for class 1 and blue for class 2. The learned potentials for $u_\theta(x)$, $u_\theta(x, y = 1)$ and $u_\theta(x, y = 2)$ are shown in (b)(c)(d) respectively.

## 14 LATENT SPACE INTERPOLATION

Figure 3 shows that the auxiliary generator smoothly outputs transitional samples as the latent code $h$ moves linearly in the latent space. The interpolated generation demonstrates that the model has indeed learned an abstract representation of the data.

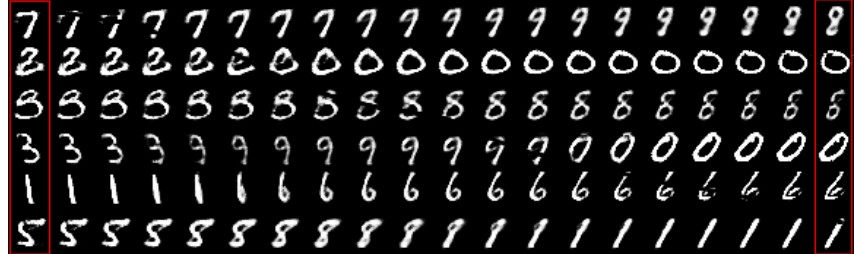

Figure 3: Latent space interpolation with inclusive-NRFs on MNIST. The leftmost and rightmost columns are from stochastic generations $x_1$ with latent code $h_1$ and $x_2$ with $h_2$. The columns in between correspond to the generations from the latent codes interpolated linearly from $h_1$ to $h_2$.

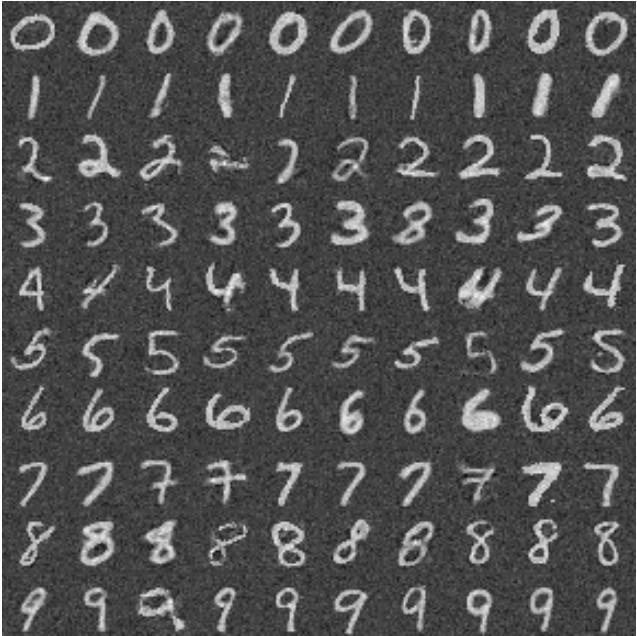

Figure 4: Conditional generated samples from semi-supervised inclusive-NRFs trained on MNIST. Due to sample revision, the background pixels are not purely black.

## 15 CLASS-CONDITIONAL GENERATION

Figure 4 shows class-conditional generation results on MNIST with semi-supervised inclusive-NRFs. Notice that the generator does not explicitly include class labels, thus it is unable to perform class-conditional generation directly. However, the random field has modeling of $p_\theta(x, y)$, based on which we can perform class-conditional generation as follows:

1. Generate a sample $x$ unconditionally, by ancestral sampling with the generator.
2. Predict the label $y$ for the sample $x$ by the random field;
3. Starting from $x$, running SGLD/SGHMC revision with $p_\theta(x|y)$ as the target density by fixing $y$. The resulting samples could be viewed as conditional generations, according to Theorem 1.

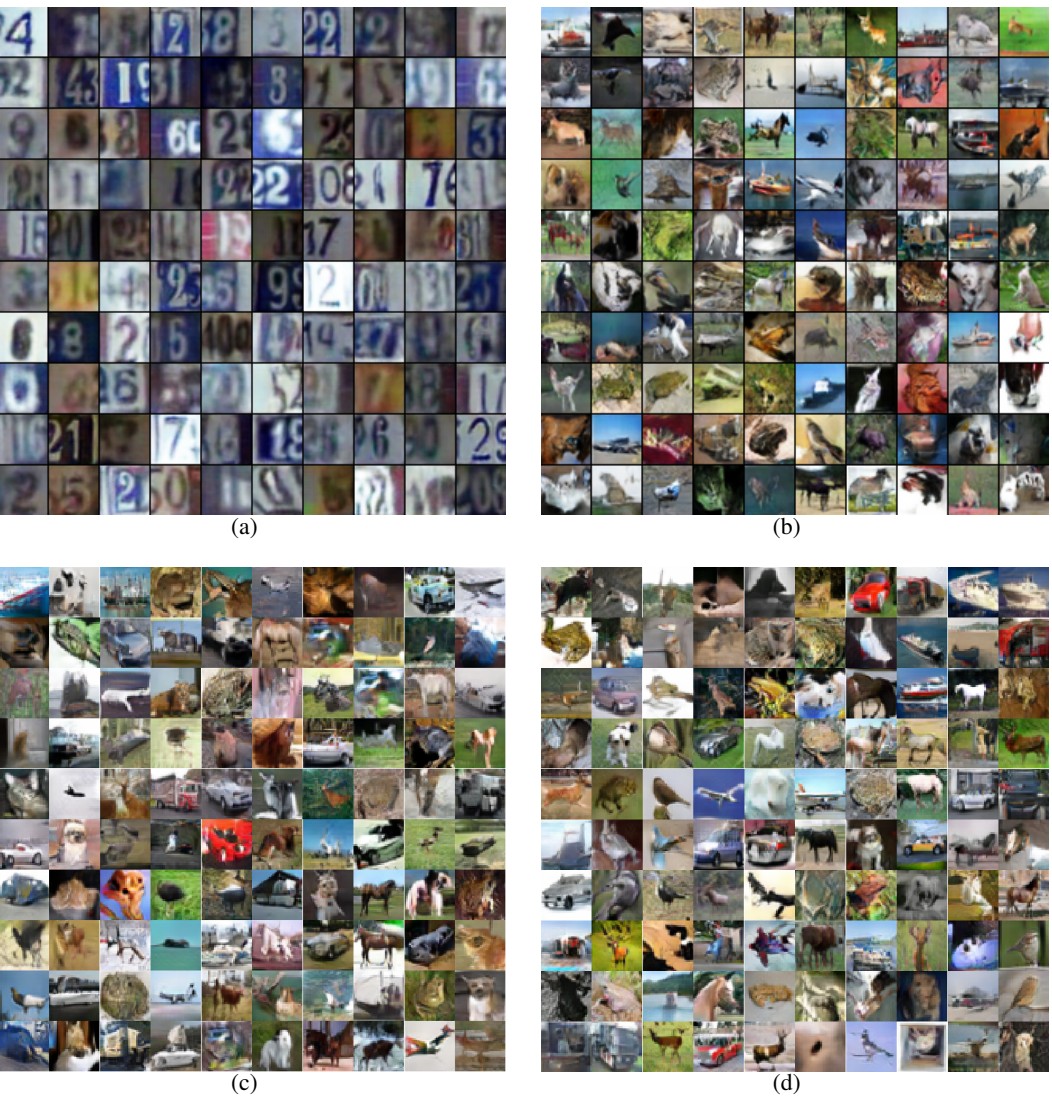

Figure 5: Generated samples from semi-supervised inclusive-NRFs (i.e. trained for SSL) on SVHN and CIFAR-10 are shown in (a) and (b) respectively. Generated samples from unsupervised and supervised training of inclusive-NRFs on CIFAR-10 are shown in (c) and (d) respectively.

Table 5: Network architectures and hyperparameters for the 2D GMM data.

| Random Field | Generator |
|---|---|
| Input 2-dim data | Noise $h$ (2-dim) |
| MLP 100 units, Leaky ReLU | MLP 50 units, ReLU |
| MLP 100 units, Leaky ReLU | MLP 50 units, ReLU |
| MLP 1 unit, Linear | MLP 2 units, Linear |
| Batch size | 100 |
| Number of iterations | 160,000 |
| Leaky ReLU slope | 0.2 |
| Learning rate | 0.001 |
| Optimizer | Adam ($\beta_1 = 0.5, \beta_2 = 0.9$) |
| Sample revision steps | $L = 10$ |

Table 6: Network architectures and hyperparameters for semi-supervised inclusive-NRFs on MNIST

| Random Field | Generator |
|---|---|
| Input $28 \times 28$ Gray Image | Noise $h$ (100-dim) |
| MLP 1000 units, Leaky ReLU, Weight norm | MLP 500 units, Sotfplus, Batch norm |
| MLP 500 units, Leaky ReLU, Weight norm | MLP 500 units, Sotfplus, Batch norm |
| MLP 250 units, Leaky ReLU, Weight norm | MLP 784 units, Sigmoid |
| MLP 250 units, Leaky ReLU, Weight norm | |
| MLP 250 units, Leaky ReLU, Weight norm | |
| MLP 10 units, Linear, Weight norm | |
| Batch size | 100 |
| Number of epochs | 200 |
| Leaky ReLU slope | 0.2 |
| Learning rate | 0.001 |
| Optimizer | Adam ($\beta_1 = 0.0, \beta_2 = 0.9$) |
| Sample revision steps | $L = 20$ |
| $\alpha$ in SSL | $\alpha_d = 10, \alpha_c = 10, \alpha_p = 0$ |

Table 7: Network architectures and hyperparameters for semi-supervised inclusive-NRFs on SVHN

| Random Field | Generator |
|---|---|
| Input $32 \times 32$ Colored Image | Noise $h$ (100-dim) |
| $3 \times 3$ conv. 64, Leaky ReLU, Weight norm | MLP 8192 units, ReLU, Batch norm |
| $3 \times 3$ conv. 64, Leaky ReLU, Weight norm | Reshape $512 \times 4 \times 4$ |
| $3 \times 3$ conv. 64, Leaky ReLU, Weight norm | $5 \times 5$ deconv. 256, ReLU, Stride=2 |
| stride=2, dropout2d=0.5 | $5 \times 5$ deconv. 128, ReLU, Stride=2 |
| $3 \times 3$ conv. 128, Leaky ReLU, Weight norm | $5 \times 5$ deconv. 3, Tanh, Stride=2 |
| $3 \times 3$ conv. 128, Leaky ReLU, Weight norm | |
| $3 \times 3$ conv. 128, Leaky ReLU, Weight norm | |
| stride=2, dropout2d=0.5 | |
| $3 \times 3$ conv. 128, Leaky ReLU, Weight norm | |
| $1 \times 1$ conv. 128, Leaky ReLU, Weight norm | |
| $1 \times 1$ conv. 128, Leaky ReLU, Weight norm | |
| MLP 10 units, Linear, Weight norm | |
| Batch size | 100 |
| Number of epochs | 400 |
| Leaky ReLU slope | 0.2 |
| Learning rate | 0.001 |
| Optimizer | Adam ($\beta_1 = 0.0, \beta_2 = 0.9$) |
| Sample revision steps | $L = 10$ |
| $\alpha$ in SSL | $\alpha_d = 10, \alpha_c = 10, \alpha_p = 0$ |

Table 8: Network architectures and hyperparameters for semi-supervised inclusive-NRFs on CIFAR-10

| Random Field | Generator |
|---|---|
| Input $32 \times 32$ Colored Image | Noise $h$ (100-dim) |
| $3 \times 3$ conv. 128, Leaky ReLU, Weight norm | MLP 8192 units, ReLU, batch norm |
| $3 \times 3$ conv. 128, Leaky ReLU, Weight norm | Reshape $512 \times 4 \times 4$ |
| $3 \times 3$ conv. 128, Leaky ReLU, Weight norm | $5 \times 5$ deconv. 256, ReLU, Stride=2 |
| stride=2, dropout2d=0.5 | $5 \times 5$ deconv. 128 ReLU, stride=2 |
| $3 \times 3$ conv. 256, Leaky ReLU, Weight norm | $5 \times 5$ deconv. 3, Tanh, Stride=2 |
| $3 \times 3$ conv. 256, Leaky ReLU, Weight norm | |
| $3 \times 3$ conv. 256, Leaky ReLU, Weight norm | |
| stride=2, dropout2d=0.5 | |
| $3 \times 3$ conv. 512, Leaky ReLU, Weight norm | |
| $1 \times 1$ conv. 256, Leaky ReLU, Weight norm | |
| $1 \times 1$ conv. 128, Leaky ReLU, Weight norm | |
| MLP 10 units, Linear, Weight norm | |
| Batch size | 100 |
| Number of epochs | 600 |
| Leaky ReLU slope | 0.2 |
| Learning rate | 0.001 |
| Optimizer | Adam ($\beta_1 = 0.0, \beta_2 = 0.9$) |
| Sample revision steps | $L = 10$ |
| $\alpha$ in SSL | $\alpha_d = 100, \alpha_c = 0, \alpha_p = 0.1$ |

