# OpenReview forum: "Learning Neural Random Fields with Inclusive Auxiliary Generators"
_ICLR.cc/2019/Conference_

### Official Review · AnonReviewer1 · 2018-11-01
**Incremental Contribution**

**Rating:** 5
**Confidence:** 3

**Review:**

The paper proposes the inclusive neural random field model. Compared the existing work, the model is different because of the use of the inclusive-divergence minimization for the generative model and the use of stochastic gradient Langevin dynamics (SGLD) and stochastic gradient Hamiltonian Monte Carlo  (SGHMC) for sampling. Experimental results are reported for unsupervised, semi-supervised, and supervised learning problems on both synthetic and real-world datasets. Specific comments follow:

1. A major concern of the reviewer is that, given the related work mentioned in Section 3, whether the proposed method exerts substantial enough contribution to be published at ICLR. The proposed method seems like an incremental extension of existing works.

2. A major claim by the authors is that the proposed techniques can help explore various modes in the distribution. However, this claim can only seem easily substantiated by experiments on synthetic data. It is unclear whether this claim is true in principle or in reality.

Other points:
3. the experimental results of the proposed method seems marginally better or comparable to existing methods, which call in question the necessity of the proposed method.

4. more introduction to the formulation of the inclusive-divergence minimization problem could be helpful. The presentation should be self-contained.

5. what makes some of the statistics in the tables unobtainable or unreported?


============= After Reading Response from Authors ====================

The reviewer would like to thank the authors for their response. However, the reviewer is not convinced by the authors’ argument.

“The target NRF model, the generator and the sampler are all different.”
It is understandable that modeling continuous data can be substantially different from modeling discrete data. Therefore, it is non-surprising that the problem formulations are different.

As for SGLD/SGHMC and the corresponding asymptotic theoretical guarantees, this reviewer agrees with reviewer 2’s perspective that it is a contribution made by this paper. But this reviewer is not sure whether such a contribution is substantial enough to motivate acceptance.

The explanation for better mode exploration of the proposed method given by the authors are the sentences from the original paper. The reviewer is aware of this part of the paper but unconvinced.

In terms of experiments, sample generation quality seems to be marginally better. Performances in multiple learning settings are comparable to existing methods.

A general advice on future revision of this paper is to be more focus, concrete, and elaborative about the major contribution of the paper. The current paper aims at claiming many contributions under many settings. But the reviewer did not find any of them substantial enough.

---

> ### Author Response · Authors · 2018-11-18
> **Author Response**
>
> > A major concern of the reviewer is that, given the related work mentioned in Section 3, whether the proposed method exerts substantial enough contribution to be published at ICLR. The proposed method seems like an incremental extension of existing works.
>
> Please refer to the above for our novelty and contribution.
>
> > A major claim by the authors is that the proposed techniques can help explore various modes in the distribution. However, this claim can only seem easily substantiated by experiments on synthetic data. It is unclear whether this claim is true in principle or in reality.
>
> This claim is empirically validated and conceptually plausible by the following reasoning. By the property of minimizing inclusive-divergence, the generator tends to cover modes of the target density $p_\theta$. The SGLD/SGHMC sampling further pushes the samples towards the modes of $p_\theta$. Presumably, this helps to produce Markov chains that mix fast between modes and facilitate model learning.
>
> > the experimental results of the proposed method seems marginally better or comparable to existing methods, which call in question the necessity of the proposed method.
>
> In addition to the strong experimental results of inclusive-NRFs, one fundamental benefit of our inclusive-NRF approach is that, unlike in GANs, we can learn density estimate about the data manifold (partly illustrated in Figure 1).
>
> > more introduction to the formulation of the inclusive-divergence minimization problem could be helpful. The presentation should be self-contained.
>
> Thanks for your suggestion. We expand section 2.1 and provide the new Proposition 1 in the Supplement.
>
> > what makes some of the statistics in the tables unobtainable or unreported?
>
> Thanks for your suggestion. We update Table 3 to differentiate the two cases - 1) the results are not reported in the original paper and without released code; 2) not applicable, e.g. the models cannot generate samples stochastically.

---

> ### Author Response · Authors · 2018-11-27
> **Author Response to Reviewer-1 feedback**
>
> Thank you very much for taking time to read our response and give feedback.
>
> We are encouraged by your acknowledgement of contributions made by this paper - noticing that modeling continuous data can be substantially different from modeling discrete data, and theoretical guarantees for applying SGLD/SGHMC.
>
> > The explanation for better mode exploration of the proposed method given by the authors are the sentences from the original paper. The reviewer is aware of this part of the paper but unconvinced.
>
> Better mode exploration of the proposed method is experimentally demonstrated in the paper, especially in the GMM experiment (Figure 1 and Table 1), as also commented by both Reviewer 1 - "Experiments are sufficient and convincing, especially the synthetic data experiments with GMM distributions." and Review 2 - "The toy example with mixture of gaussians is convincing showing the contrast in results between the exclusive NRF, inclusive, and sampling gradient revision steps.".
>
> Our explanation for this improvement is mainly based on different properties between inclusive and exclusive divergences, which is commonly acknowledged when discussing the issue of mode exploration.
>
> > In terms of experiments, sample generation quality seems to be marginally better. Performances in multiple learning settings are comparable to existing methods.
>
> Besides the strong experimental results (better/on par with state-of-the-art), one fundamental additional benefit of our inclusive-NRF approach is that, unlike in GANs, it enables us to learn density estimate about the data manifold (partly illustrated in Figure 1).
>
> > A general advice on future revision of this paper is to be more focus, concrete, and elaborative about the major contribution of the paper. The current paper aims at claiming many contributions under many settings. But the reviewer did not find any of them substantial enough.
>
> Please read our updated response for novelty and contribution above after receiving your feedback to our initial response.
>
> Examinations under multiple learning settings are better viewed as a plus rather than a minus, which elaborates our major contribution - the proposed inclusive-NRF approach, as we explained in our response above.

---

### Official Review · AnonReviewer3 · 2018-11-02
**appears effective, though unfamiliar with this type of model**

**Rating:** 6
**Confidence:** 2

**Review:**

This paper describes a method of training NRFs with auxiliary generator networks, using an error that minimizes KL(NRF || generator).  This formulation enables the use of iterative gradient-based stochastic sampling of image samples from the model distribution using SGLD/SGHMC.  Applications to both unsupervised sample generation and semi-supervised classification are evaluated.

I'm not very familiar with these types of NRFs or random sampling techniques, but the approach appears sound and is evaluated rather well.  I would have liked some more background and explicit description and contrast compared to the explicit NRF.  While this is described already, I think the contrasts could potentially be spelled out even more explicitly, particularly in the descriptions of the sampling algorithms.

The toy example with mixture of gaussians is convincing showing the contrast in results between the exclusive NRF, inclusive, and sampling gradient revision steps.

Experimental evaluations on MNIST, SVHN and CIFAR show that the system obtains performance similar to SOA generative systems, in both semi-supervised classification and sample generation.


Questions and comments:

- While the paper claims the results show classification and generation performance are complementary, Table 3 appears to validate the opposite claim, that these are to a large degree a trade-off.  The fact that this system performs well at both is good, but to me it looks like it may be on the "shoulder" of a frontier curve if one were to plot the classification vs generation performance of the different current systems.

- Table 4 and sec 4.4:  I think these could be clearer.  The first observation states that revision improves IS.  But using more iterations (increasing L) does not appear to increase IS.  There does appear to be a consistent increase from the first column (generation) to second (revision), though -- is this what this observation refers to?  In addition, I'm not entirely clear what the "Generation IS" vs "Revision IS" column refers to --- I believe "generation" is the initial sampling of x=g(h) (i.e. h followed by q(x | h)), and "revision" is the application of gradient revision.  But then how does the generation IS results change from row to row (which only modify the revision step)?

---

> ### Author Response · Authors · 2018-11-18
> **Author Response**
>
> We appreciate that you found our paper to be effective.
>
> > While the paper claims the results show classification and generation performance are complementary, Table 3 appears to validate the opposite claim, that these are to a large degree a trade-off.  The fact that this system performs well at both is good, but to me it looks like it may be on the "shoulder" of a frontier curve if one were to plot the classification vs generation performance of the different current systems.
>
> We do not claim the results show classification and generation performance are complementary. As we comment in the footnote under section 4.3, the conflict of good classification and good generation is reported when using the (K + 1)-class GAN-like discriminator objective for SSL, and does not seem to be reported in previous generative SSL methods (Zhu (2006); Larochelle et al. (2012)) which use the K-class classifier like in semi-supervised inclusive-NRFs.
>
> What we claim is that the conflict of GAN-based SSL is embarrassing and in fact obviates the original idea of generative SSL - successful generative training, which indicates good generation, provides regularization for finding good classifiers (Zhu, 2006; Larochelle et al., 2012). In this sense, Bad-GANs could hardly be classified as a generative SSL method.
>
> The raised viewpoint of trade-off between classification and generation is interesting. Different SSL methods and models, e.g. generative model based SSL (e.g. Zhu, 2006; Larochelle et al., 2012) and discriminative SSL (e.g. Miyato et al., 2017), use different regularization. Does this trade-off exist for all SSL methods and models, and if not, for what type of SSL methods and models? Is the trade-off due to the regularization used? Exploration of such problem is interesting but outside the scope of this paper.
>
> > Table 4 and sec 4.4:  I think these could be clearer.
>
> Thanks for your suggestion. We update sec 4.4 and add explanation in the caption of Table 4.
>
> > There does appear to be a consistent increase from the first column (generation) to second (revision), though -- is this what this observation refers to?
>
> Yes.
>
> > In addition, I'm not entirely clear what the "Generation IS" vs "Revision IS" column refers to --- I believe "generation" is the initial sampling of x=g(h) (i.e. h followed by q(x | h)), and "revision" is the application of gradient revision.  But then how does the generation IS results change from row to row (which only modify the revision step)?
>
> Column-wise "Generation IS" vs "Revision IS" is to compare the two manners to generate samples  - whether applying sample revision or not in inference (generating samples) given a trained NRF, as previously illustrated in Figure 1 over synthetic GMM data. For both manners, the NRF model is trained with the sample revision step.
>
> Each row in Table 4 represents a specific setting in model training, such as using SGLD or SGHMC and the revision step $L=1/5/10$ used. Thus, each row produces a specific different NRF model.

---

### Official Review · AnonReviewer2 · 2018-11-02
**Interesting but incremental**

**Rating:** 6
**Confidence:** 3

**Review:**

This paper addresses an important problem of learning the random field using neural networks by using a inclusive auxiliary generator. Comparing to existing state-of-the-art methods for learning neural random fields, this paper used a the inclusive-divergence (KL divergence of the density approximate and the auxiliary generator) which avoids the intractable entropy term. SGLD/SGHMC are used to revise samples drawn from the auxiliary generator and these two sampling methods are examined theoretically.

In generally, the paper is well motivated and well written. Experiments are sufficient and convincing, especially the synthetic data experiments with GMM distributions.

However, I am a little bit concerned that the theoretical contribution seems weak. As discussed in the related work, the idea of using neural network to learn the random field is not new. Using inclusive-divergence is also not new, e.g. Xie et al (2016) and Wang & Ou (2017) already proposed to use the inclusive-divergence. If I understand it correctly, the only contribution here is to apply the SGLD/SGHMC to revise the samples and authors provided some theoretical analysis of SGLD/SGHMC.

The overall technical quality of the paper is sound but I am not 100% sure about the equations, e.g. the second line in Eq. 4.

In summary, this paper is well written and authors have done a good job. But I will appreciate if authors can elaborate
more on the novelty and innovation of the paper.

---

> ### Author Response · Authors · 2018-11-18
> **Author Response**
>
> We appreciate that you found our paper to be interesting. Please refer to the above for our novelty and contribution.
>
> >the idea of using neural network to learn the random field is not new.
>
> Yes but received less attention with slow progress. We significantly advance the learning of NRFs, both theoretically and empirically. Please refer to the above for our novelty and contribution.
>
> > Using inclusive-divergence is also not new, e.g. Xie et al (2016) and Wang & Ou (2017) already proposed to use the inclusive-divergence.
>
> The Related Work section in the updated paper is expanded to more clearly reveal the differences between this paper and previous studies, e.g. Xie et al (2016) and Wang & Ou (2017).  Also please refer to the above for our novelty and contribution.
>
> > The overall technical quality of the paper is sound but I am not 100% sure about the equations, e.g. the second line in Eq. 4.
>
> We add the proof for the second line of Eq. 4 in the new Proposition 1 in the Supplement.

---

### Author Response · Authors · 2018-11-18
**Author Response: for novelty and contribution**

We would like to thank all of the reviewers for their thoughtful and helpful comments. We have uploaded a new version of our manuscript with improvements based on reviewer feedback.

For the novelty and contribution of the paper:

The Related Work section in the updated paper is expanded to more clearly reveal the differences between this paper and previous studies, e.g. Xie et al (2016) and Wang & Ou (2017).

Learning in Wang & Ou (2017) and in this paper minimizes the inclusive-divergence $KL[p_\theta||q_\phi]$ w.r.t. $\phi$.  But noticeably, this paper presents our innovation in development of NRFs for continuous data, which is fundamentally different from Wang & Ou (2017) for discrete data. The target NRF model, the generator and the sampler are all different.
Wang & Ou (2017) mainly studies random field language models, using LSTM generators (autoregressive with no latent variables) and employing Metropolis independence sampler (MIS) - applicable for discrete data (natural sentences).
In this paper, we mainly develop random field models for continuous data (images), using latent-variable generators and utilizing SGLD/SGHMC (with solid theoretical examination) to exploit gradient information in the continuous space.

In Xie et al (2016), motivated by interweaving maximum likelihood training of the random field $p_\theta$ and the latent-variable generator $q_\phi$, a joint training method is introduced. Operationally, in learning $\theta$ and $\phi$, this method also uses Langevin sampling to generate samples. Two Langevin sampling steps are intuitively interleaved according to gradients w.r.t. x and h separately. This is different from our sampling step, which moves $(h,x)$ jointly, as theoretically justified in section 2.2. Moreover, interpretation presented in Xie et al (2016) relates their method to a joint optimization problem, which is also different from ours as shown in Eq. (4). Thus, learning in Xie et al (2016) does not aim to minimize the inclusive-divergence $KL[p_\theta||q_\phi]$ w.r.t. $\phi$. [We correct our misunderstanding of Xie et al (2016) based on its recent version. See our updated paper for details.]

Please refer to the updated related work for more comparisons. It can be seen that there are non-trivial differences in model formulation, algorithm development, and theoretical examination between this paper and previous studies. This paper makes significant contribution in learning NRFs, instead of incremental contribution, as explained below.

In a word, our major contribution/our major claim is:  we propose the inclusive-NRF approach for continuous data, with _solid theoretical examination_ on exploiting gradient information in model sampling and with _extensive strong experimental results_ compared to the state of the art.
* The major contribution is substantiated by the following developments, which are significantly new and have never been reported/obtained before:
 - The Eq. (5) in section 2.1 with Proposition 1 in the Supplement (model formulation);
 - The whole section 2.2 (solid theoretical examination on applying SGLD/SGHMC);
 - The whole section 2.3 (SSL with inclusive-NRFs).
* We show that inclusive-NRFs can be flexibly used unsupervised/supervised image generation and semi-supervised classification, and empirically to the best of our knowledge, represent the best-performed random fields in these tasks.
* Extensive empirical evaluations show that inclusive-NRFs obtain state-of-the-art sample generation quality and achieve strong semi-supervised learning results on par with state-of-the-art DGMs.

Specific responses to each reviewer are provided in the following. We are happy to discuss any questions that you may have during the discussion period.

---

### Author Response · Authors · 2018-11-29
**We make substantial contribution, instead of incremental contribution.**

Dear Reviewers,

Thank you again for your valuable comments and for considering our responses and revisions. The main revisions of the manuscript  are: revising the Abstract, expanding the Related Work section to more clearly reveal the differences between this paper and previous studies (to respond to reviewer 1 & 2), adding the new Proposition 1 in the Supplement to prove the old Eq. 4 (to respond to reviewer 2), updating sec 4.4 and the caption of Table 4 to make them clearer (to respond to reviewer 3).

Basically, we do not think the comment on our lack of substantial contributions is reasonable, considering the contributions clarified below [Please read our updated response for novelty and contribution below after receiving reviewer-1's feedback to our initial response.]. In a word, our major contribution/our major claim is:  we propose the inclusive-NRF approach for continuous data, with _solid theoretical examination_ on exploiting gradient information in model sampling and with _extensive strong experimental results_ compared to the state of the art.

In addition to the strong experimental results of inclusive-NRFs, one fundamental benefit of our inclusive-NRF approach is that, unlike in GANs, we can learn density estimate about the data manifold (partly illustrated in Figure 1).

Given reviewer-1 acknowledging that modeling continuous data can be substantially different from modeling discrete data, “The target NRF model, the generator and the sampler are all different”, and the theoretical guarantees contributed in this paper for applying SGLD/SGHMC, we hope that we have addressed your concern on our contribution.

The discussion period is coming to an end. If you have any questions or would like to provide more specific context behind your scores, we would be happy to provide feedback.

---

### Meta-Review · Area_Chair1 · 2018-12-14

**Confidence:** 3
**Recommendation:** Reject

**Metareview:**

This paper proposes a method for learning neural RFs with the inclusive-divergence minimization problem.

Reviewers generally agree that the experiments are sufficient and convincing, and that the method is evaluated well. Results are comparable with SOTA methods for image generation. The paper is reasonably well-written.

The paper is also somewhat lacking in background; most people at ICLR will not be very familiar with this learning problem. More information on the inclusive-divergence minimization problem would be helpful. A major concern of reviewers is whether novelty of the method is sufficient for publication.

---

> ### Author Response · Authors · 2018-12-21
> **We are very surprised and confused by the decision, given that “Reviewers generally agree that the experiments are sufficient and convincing, and that the method is evaluated well ... The paper is reasonably well-written.”**
>
> Regarding novelty, this paper represents the first neural random field paper achieving state of the art results. We have addressed the reviewers' concern on novelty, but received no further concrete response.
>
> Un-familiarity with the new approach should not be the reason to reject the paper. Rather, given that this learning problem is clearly under-appreciated and the proposed method is theoretically sound and empirically promising, it is more worthwhile to help promote further research in this NEW approach.
>
> With full respect, we would greatly appreciate it if the paper could be reconsidered.